# The Virtual Reality Tour: Immersive Preoperative Information for Elderly Patients

**DOI:** 10.3390/healthcare13222896

**Published:** 2025-11-13

**Authors:** Karsten Lomholt Lassen, Carina Sjöberg, Annelie Augustinsson, Maria Joost, Nanna Wagner Christiansen, Anja Geisler, Pether Jildenstål

**Affiliations:** 1Department of Health Sciences, Care in High Technological Environments, Lund University, 221 00 Lund, Sweden; carina.sjoberg@med.lu.se (C.S.); annelie.augustinsson@med.lu.se (A.A.); pether.jildenstal@vgregion.se (P.J.); 2Department of Anesthesiology, Zealand University Hospital, 4600 Køge, Denmark; majoo@regionsjaelland.dk (M.J.);; 3Department of Urology, Zealand University Hospital, 4000 Roskilde, Denmark; 4Department of Digestive Diseases, Transplantation and General Surgery, Copenhagen University Hospital, Rigshospitalet, 2100 København Ø, Denmark; anja.edith.geisler@regionh.dk; 5Faculty of Health and Medical Sciences, Department of Clinical Medicine, Copenhagen University, 2100 København Ø, Denmark; 6Department of Anaesthesiology and Intensive Care, Skåne University Hospital, 221 00 Lund, Sweden; 7Institute of Health and Care Sciences, Sahlgrenska Academy, University of Gothenburg, 405 30 Gothenburg, Sweden; 8Faculty of Nursing and Health Sciences, Nord University, 8049 Bodö, Norway; 9Department of Anaesthesiology, Surgery and Intensive Care, Sahlgrenska University Hospital, 413 45 Gothenburg, Sweden; 10Department of Anaesthesiology and Intensive Care, Örebro University Hospital and School of Medical Sciences, Örebro University, 701 85 Örebro, Sweden

**Keywords:** anesthesia, elderly patients, patient education, preoperative information, qualitative research, total knee arthroplasty, virtual reality

## Abstract

**Background/Objectives:** Older patients are more susceptible to the process of understanding and retrieving information. To provide adequate preoperative information, reduce their vulnerability and anxiety, and thereby facilitate pre-, intra-, and postoperative care, the adoption of new information technologies is essential. This study aimed to explore elderly patients’ experiences and perceptions of a Virtual Reality (VR)-based preoperative informational tool designed for individuals scheduled to undergo TKA. **Method:** A qualitative content analysis was conducted from February to June 2025 based on 14 semi-structured post-discharge interviews. Participants were recruited as part of a randomized controlled trial. **Results:** One main theme and two categories describe patients’ experiences of using VR information, which is perceived as a valuable supplement to standard preoperative information. The category “Using VR as an information tool” includes how patients experienced the immersive environment, how it affected them, and their views on using the VR glasses. The category “Apply the information provided” described how patients evaluated and applied the VR content in relation to their expectations and the actual surgical experience. **Conclusions:** Immersive virtual reality (VR) shows promise as a preoperative information tool, improving patients’ understanding of the perioperative process and supporting engagement in their own care. Its effectiveness depends on reliable technical performance and adaptation to individual needs. VR should complement, not replace, communication with healthcare professionals. Further research is needed to identify optimal timing for VR delivery and its impact on preoperative anxiety and patient experience.

## 1. Introduction

Total knee arthroplasty (TKA) is one of the most commonly performed elective orthopedic procedures worldwide, driven by the aging population and the increasing demand for a longer active life [1]. In Denmark alone, more than 10,000 primary knee arthroplasties are performed annually, and the numbers are expected to rise due to demographic shifts and increasing life expectancy [2]. Internationally, similar trends are observed; for instance, in the United States, the annual volume of TKA is projected to exceed 1.5 million by 2030 [3]. The primary patient population consists of older adults, often with multiple comorbidities, making preoperative optimization and education particularly critical for this group [4]. Older patients often face unique challenges in understanding complex preoperative information due to factors such as cognitive decline, hearing or visual impairments, and unfamiliarity with medical terminology [5]. This heterogeneous patient group includes many individuals who undergo repeated procedures or bilateral knee surgery. Consequently, variations in information needs are influenced by patients’ social context, level of health literacy, and previous experiences [6,7]. Traditional methods of information delivery, such as written materials or verbal consultations, may be insufficient to ensure adequate comprehension and retention in this demographic [8,9,10]. Consequently, patients may undergo surgery with unrealistic expectations, elevated anxiety levels, or limited understanding of the perioperative process [9,10,11], which can place additional strain on healthcare resources and reduce overall system efficiency [12]. Patient satisfaction and perceived surgical success have been shown to correlate strongly with the degree to which preoperative expectations are fulfilled [13]. Ensuring that patients are adequately informed and mentally prepared for surgery contributes to improved emotional well-being and better clinical outcomes, including reduced postoperative pain, shorter hospital stays, lower complication rates, and faster functional recovery [14,15].

Virtual Reality (VR) has emerged as a promising technological tool, offering immersive environments that can support patients’ mental preparation for surgery [16,17].

VR technology enables users to experience complete immersion within an artificial environment featuring interactive visual, auditory, and motor domains. By presenting three-dimensional multimedia that can be manipulated and delivered through head-mounted displays and noise-reducing headphones, VR enables patients to engage with a controlled, interactive representation of the perioperative environment—such as the operating theater, preoperative waiting area, and post-anesthesia care unit (PACU)—with the potential to foster psychological readiness and a sense of reassurance before surgery [18,19,20,21]. Although both written and VR-based methods aim to inform and prepare patients, they differ in their mode of delivery, cognitive load, and potential emotional impact. Traditional methods are static and require active interpretation, whereas VR offers a multimodal learning experience that may reduce cognitive demands by presenting information in a contextualized visual format [18,21,22]. Cybersickness is a notable limitation of VR, which is affected by technology, content, and duration of exposure. However, further research is needed to understand better the impact of user-related factors, such as age and gender [23]. Given that older individuals are not typically familiar with immersive digital technologies, there is a particular need to explore how they engage with and respond to VR-based information. For this group, receiving clear and understandable preoperative information may represent a significant opportunity to regain a sense of control in an otherwise unfamiliar clinical setting. Feeling informed and capable may further contribute to an enhanced patient’s sense of competence and encourage active involvement in their care. Such experiences resonate strongly with the principles of person-centered care, which emphasize the value of acknowledging patients’ capabilities. Despite the increasing use of immersive digital technologies, such as VR, in healthcare, research exploring elderly patients’ experiences with these tools for preoperative information remains limited.

While previous studies have examined the use of VR for patient education in various contexts [24], most have primarily focused on quantitative outcomes, such as anxiety and knowledge scores [25,26]. However, there remains limited research exploring the experiences of patients scheduled to undergo TKA in receiving preoperative information through immersive VR environments.

Understanding how elderly patients perceive and engage with VR in the surgical pathway is crucial, particularly in procedures associated with anxiety and extensive rehabilitation, including TKA. Therefore, this study aimed to explore elderly patients’ experiences and perceptions of a VR-based preoperative informational tool designed for individuals scheduled to undergo TKA.

## 2. Method

### 2.1. Study Design

A qualitative design was chosen to explore patients’ subjective experiences and meanings related to using VR as preoperative information. A qualitative approach enabled an understanding of perceived usefulness, realism, and emotional responses beyond numerical outcomes [27].

Data were collected through individual, semi-structured interviews. A qualitative content analysis, as described by Graneheim and Lundman [24], was used to identify patterns, variations, differences, and similarities in the participants’ perceptions. This study followed the Consolidated Criteria for Reporting Qualitative Research (COREQ) guidelines [28].

### 2.2. The Settings and Participants

The study was conducted at a public anesthetic department in Denmark, where pre-anesthesia assessment before TKA surgery is routinely performed as part of the standardized clinical care pathway. Participants were recruited from the VR intervention arm of the ongoing randomized controlled trial. A total of 14 patients were included for individual interviews using convenience sampling from the intervention group in the randomized controlled trial (RCT; registered at ClinicalTrials.gov: NCT06292663).

Patients allocated to the intervention group received a VR experience that illustrated their patient’s journey during the scheduled preoperative assessment at the hospital, one to two weeks before surgery. The VR session lasted approximately eight minutes and was delivered via a PICO G3 4K (Qingdao, China) head-mounted display (HMD) in a quiet consultation room. A trained healthcare professional was present to introduce the equipment, assist with setup, and address any immediate questions.

The VR content was developed in partnership with the Danish technology firm KHORA (Copenhagen, Denmark) [29], providing a guided and immersive walkthrough of the patients’ surgical journey from admission to postoperative recovery. The content presents the patient’s journey on the day of surgery from a first-person perspective, recorded with a 360° camera. The film follows a patient entering the ward, meeting the nurse, entering the operating theatre (excluding footage of the surgery itself), awakening in the recovery area, and ending when the patient returns to the ward. The narrative is voice-guided with natural ambient sound, filmed under standard lighting conditions from a standing viewpoint to simulate the patient’s perspective. All scenes were filmed using natural clinical lighting and edited in sequential order, without the addition of any effects. The narrative followed the chronological flow of the surgical day.

Figure 1 provides example screenshots of the content.

Patients were eligible for inclusion if they were scheduled for TKA under spinal anesthesia, aged 18 years or older, capable of understanding the study protocol, including its potential risks and benefits, and provided written informed consent. Patients were excluded if they were unable to read or understand Danish, were deemed uncooperative by the investigators, suffered from claustrophobia or related anxiety disorders, had conditions preventing the use of VR equipment, or had significant visual or auditory impairments that could interfere with the VR experience. The characteristics of the participants are presented in Table 1.

### 2.3. Data Collection

Participants were contacted after hospital discharge and invited to participate in the qualitative study. As part of the initial trial enrollment process, all participants were informed that they might be approached for a follow-up qualitative interview to explore their experiences with the intervention in the future. The study was explained both orally and in a written informed consent form. All interviews were subsequently conducted via telephone within two to four weeks postoperatively to capture recent experiences.

A semi-structured interview guide was used based on research on whether VR could be considered part of the surgical pathway [30]. The guide included open-ended questions beginning with the question “Can you describe your thoughts on the 360° VR experience you saw before the surgery?” All questions were designed to explore participants’ perceptions of VR-based preoperative information, their emotional and cognitive responses to the technology, how the information aligned with their experience, and the perceived usefulness of the intervention in preparing for surgery and recovery. K.L.L. and M.J. conducted all interviews, which were audio-recorded using an encrypted, password-protected tablet device, and later transcribed by Viceron.com (Copenhagen, Denmark). Interviews were conducted by telephone at times chosen by the participants, in settings they selected themselves, all of which were in their homes.

Telephone interviews, which lasted a median of 10.5 min (range 6.5–16 min), were conducted between February and June 2025, with a median of 12 days (range 7–23 days) post-discharge. Data saturation was considered reached when at least two consecutive interviews produced no new codes or subcategories relevant to the research question [27].

The number of participants included in this study appears sufficient, particularly as the sample represents a homogenous group within a narrowly defined research topic [31].

### 2.4. Data Analysis

Interview recordings were transcribed using Viceron.com. To ensure accuracy, all transcribed recordings were verified by the researchers against the original audio recordings. When the text was imprecise due to the use of the AI Tool, the text was corrected accordingly. All interviews were.

The transcribed text was systematically analyzed using a qualitative content analysis approach [32]. The analysis focused on both manifest and latent content, starting with each transcript being read multiple times to gain a comprehensive understanding of its content. CS and KL were mainly responsible for the analysis. NVivo v.14.24.3 (Denver, CO, USA) was used to extract 92 meaning units of the text relevant to the research aim, which were identified and condensed while preserving their core meanings. The condensed meaning units were labeled with 12 codes. In the analysis, codes were compared and grouped into six subcategories, which were then abstracted into two broader categories, as described by Graneheim and Lundman [32]. The main theme, latent content, describes the phenomena. Examples of data analysis are listed in Table 2.

## 3. Ethical Considerations

The study received ethical approval from The Region Zealand, Approval No: EMN-2023–08767. This study was conducted in accordance with the Declaration of Helsinki, the General Data Protection Regulation (GDPR), and applicable national data protection laws. All participants were informed about the study and provided written and audio-recorded consent prior to their participation.

## 4. Results

The Main Theme, “Individualized VR information with VR means additional value to the patient experience,” is illustrated by the categories “Using VR as an information tool” and “Apply the information provided,” with subcategories, presented in Table 3. Describing patients’ experiences of receiving preoperative information through VR head-mounted devices in relation to their TKA.

### 4.1. Using VR as an Information Tool

This category describes the experience of wearing glasses and engaging in immersive activities, as well as their impact on the patient’s emotions.

### 4.2. Adapting to VR Glasses

Patients generally described their experience of using VR glasses as novel and engaging. Well, that was the first time I tried it. Wearing a pair of glasses like that, it seemed like you were in the operating room and on the way in, and all that. Yeah, it was very lifelike (Patient 10).

For many, this was their first encounter with VR technology, and their reactions ranged from fascination to initial discomfort. Most patients quickly adapted to the unfamiliar equipment, with one noting, “Of course, it felt strange with those glasses, but I got used to it very quickly” (P4). However, several challenges that impact the overall experience have been described. Some patients noted technical limitations, including issues with sound synchronization and the need to physically turn their head or rotate their body to follow visual content. This occasionally resulted in mismatches between the audio and visuals, requiring patients to adjust their position to engage with the 360-degree view fully. These challenges led some patients to express skepticism about the technology, suggesting that the same content may have been equally effective when presented on a standard screen. Although these technical issues did not dominate the overall impression, they were described as minor annoyances that slightly interfered with the intended immersive experience. Despite this, the patients were able to see the benefits and potential.

### 4.3. Take Part in an Immersive Experience

This category reveals how patients experienced the immersive experience of the VR system. Several patients described the sensation of “being there” in the hospital environment, described as “Very realistic, like being in the operating room and on the way in” (P10).

Visual and auditory elements were appreciated for making the information more engaging and easier to understand. Patients emphasized that detailed imagery helped them retain content more effectively than traditional paper-based formats. For many, the experience was described as either positive or exciting. The immersive experience was highly valued and perceived as educational and therefore considered useful by the patients. The VR-based information influenced the patients’ emotional state before surgery. While some patients reported feeling calm or unconcerned from the outset, others described how the video contributed to their reassurance and reduced their anxiety. Several patients indicated that the VR experience helped them feel more at ease and secure by providing familiarity with the clinical setting. “Yes, it’s like you’re being held by the hand the whole way through” (P5). Additionally, the impact of the VR experience could also serve to de-dramatize patients’ expectations of how the procedure was expected to be carried out. Patients also reported that the VR content helped alleviate their fears or clarify previous concerns. The visual content provided more prosperous and diverse information than traditional written information. Others, however, reported no emotional impact either because they had no pre-existing fear or because the video confirmed what they had already expected. Thus, those who were not nervous or emotionally affected before the procedure experienced no differences. Although participants were asked, according to the interview guide, whether the VR experience affected their expectations regarding pain during the surgical process, none spontaneously reported any influence on their perception of pain.

### 4.4. Apply the Information Provided

This category describes the perception when the VR content did not match the real experience. Additionally, how the content met expectations before surgery influenced the compliance and understanding of the process.

### 4.5. Discrepancies Between VR and Clinical Experience Are Annoying

Patients reacted when the VR-based information was not fully aligned with their real-world clinical experience. Although the information was not perceived as intentionally misleading, some patients felt it was inaccurate. While a few considered the discrepancies to be minor and unimportant, others found them misleading or frustrating, which affected their perception of the technology. Patients commented that the VR film presented a “too polished” version of the surgical experience. They expressed that they missed important procedural details, such as variations in spinal anesthesia, and that it may take several attempts to succeed with spinal anesthesia.

“They should inform patients more that sometimes it doesn’t work the first time” (P13).

Patients noted that the clinical pathway they encountered on the day of surgery did not always correspond precisely with that shown in the VR simulation. Variations included differences in the number of staff members present during specific stages or where specific procedures were performed. “For example, they weren’t as good at hitting my spine or spinal cord compared to the Virtual Reality version and things like that.” (P8). However, these discrepancies were not experienced as problematic and did not negatively influence patients’ overall sense of preparedness or trust in the care they received.

### 4.6. Influence on Expectations and Compliance

This category describes how the VR-based information influenced patients’ expectations regarding the entire surgical procedure and their ability to comply with instructions. For several patients, the VR experience primarily confirmed their existing expectations. For them, nothing changed; they received confirmation, which illustrates how the VR content validated their existing expectations. Others described a better sense of procedural clarity, which helped them feel more secure and ready to comply with instructions: “Yes, yes, I knew I had to curl up and those things—so yes, it was fine” (P2). A few patients found the information less impactful, reporting that their expectations remained unchanged or that there was a mismatch due to prior negative experiences. Some noted that they would have preferred more detailed information about the surgical procedure. Overall, their influence on expectations and compliance ranged from reassurance and confirmation to a perceived lack of added value. They emphasized that the use of precise language without reliance on medical jargon was highly beneficial. Patients described how the VR experience contributed to their understanding of the procedural tasks, thereby increasing their sense of deeper understanding. For many, the visualization of the patient journey provided clarity and confirmed their existing understanding of the tasks that would take place on the day of surgery. It also helped them prepare for participation and decision-making, in which they could be involved. Patients found it interesting to prepare for what was going to be an actively involved experience. This was perceived as an advantage in understanding the process step by step. “It gave me an insight into what was going to happen, and the process—I was happy about that” (P1). For some patients, the VR session confirmed what they already knew, whereas others described it as adding a new dimension to their preparation. The experience was that it was essential to gain insight into the process, observe different environments, and develop an understanding of how it works.

## 5. Discussion

The main finding of this study was that a contextualized visual format for information, combined with VR, provides added value to patients. This is particularly true when tailored to meet each patient’s specific information needs. This type of presentation offers clarity on aspects that are often overlooked in traditional written and oral information delivery. Particularly valuable is the immersive format, which engages multiple senses featuring interactive visual, auditory, and motor domain elements [30].

The head-mounted display was generally perceived as easy to use and well accepted by the patients in this study. In line with a scoping review [33], patients consider VR beneficial for improving their understanding of medical information. Schubbe et al. [34] conducted a systematic review and meta-analysis, demonstrating that the use of pictorial health information enhances knowledge and understanding among patients and consumers, particularly benefiting individuals with lower health literacy. Patients in the current study highlighted the added value of visuals and found the pre-information video more detailed and easier to comprehend. Many people are visual learners, which may explain the positive effects of VR [35]. None of the patients experienced cybersickness despite using fully immersive VR. Nevertheless, it remains essential that traditional information materials are available for patients who are unable or unwilling to engage in VR technology for various reasons. Some problematic occasions and technical issues required patients to turn their heads to view the content fully, which was bothersome. Although this issue was resolved, it underscores the importance of technical reliability and the significance of healthcare professionals’ attitudes. In a study of multidisciplinary hospital staff, Shiner et al. [36] found that despite limited previous VR experience, the healthcare professionals expressed positive attitudes toward VR but noted barriers such as inadequate IT support, limited technical skills, and time constraints.

The immersive experience allowed patients to construct knowledge from their own perspective through self-driven, interactive processes. Since patients often have a limited and varied understanding of the perioperative pathway, VR helped address their individual needs. For some, it was valuable to gain insight into the roles of different healthcare professionals and the timing of their involvement. For those who had undergone the same surgery previously, the experience served as an update or confirmation, whereas for first-time patients, it provided entirely new information. Some patients were already calm and without worries, but still appreciated the VR experience, which made them feel equally or even more secure. These findings align with van der Linde-van den Bor’s study [6] that examined patients’ information needs and perceptions of VR in preoperative patient education.

Many patients expressed worries and concerns regarding the procedure. The immersive experience helped de-escalate and calm them, reduce anxiety, and foster a sense of security and positivity, as they knew what would happen and what was expected of them. Previous studies have also demonstrated that VR education reduces anxiety and fear, while improving patient satisfaction across various healthcare settings, yielding positive results [17,24]. Insights into what would happen and what would be expected helped reduce the asymmetrical relationship between patients and healthcare professionals. With greater knowledge gained through VR, patients in the current study perceived that they could participate more actively in their care and decision-making. Vogel et al. [37] identified several barriers to shared decision-making. To overcome these challenges, patients must be recognized as having the competence and authority to engage meaningfully in the process. None of the patients in this study reported that VR affected their perioperative pain experience. This aligns with the findings of a randomized controlled trial by Bekelis et al. [18], in which patients who were exposed to a preoperative immersive VR experience during the perioperative period reported greater satisfaction, improved preparedness, and reduced stress. However, no differences were observed in pain levels.

Many patients felt that VR presented an idealized version of the process that did not reflect potential difficulties. For example, more healthcare staff were present during the actual procedure than were depicted in the VR experience. Several patients reported experiencing multiple attempts before spinal anesthesia was successfully administered, which left them feeling unprepared because the VR scenario depicted only a single successful attempt. One possible way to address this and accommodate individual information needs could be a solution, such as that used in the study by van Rijn et al. [24], where patients were given the option to access additional information about specific rooms or procedures within the VR environment if they wished.

In contrast, VR content met patient expectations positively. The use of explicit language and the absence of medical terms were considered beneficial. Improvements, such as a better understanding of the positioning required during spinal anesthesia, have contributed to increased compliance. However, the perceptions of those with previous negative experiences remained unchanged. The complexity of patients’ heterogeneous information needs, influenced by personal preferences and previous hospital experiences, is confirmed in a study by van der Linde-van den Bor et al. (2022) [6].

The primary advantage of VR is its immersive experience, which fosters emotional engagement, a quality often lacking in traditional information methods. Patients in this study also highly valued the opportunity to gain insight into the process, as it helped them prepare for situations in which they could actively participate and make informed choices. From a clinical perspective, VR can be a valuable supplement to existing preoperative education methods.

### Methodological Considerations

The current study aimed to investigate elderly patients’ experiences and perceptions of a preoperative informational tool with VR for patients scheduled to undergo TKA. The qualitative content analysis was a suitable method for investigating the perspectives and perceptions of patients who had used VR. To enhance the credibility, dependability, confirmability, and transferability of this study, we rigorously followed all the steps described by Granheim and Lundman [32]. The inclusion of patients did not consider whether patients had previously undergone knee replacement; 10 of 14 patients had prior experience with the procedure.

The depth of patients’ responses when reflecting on receiving information via VR in interviews may be strongly influenced by their previous experiences and knowledge. This factor accounts for the observed variation in interview duration. Although the interviews were relatively short (averaging 10 min), data saturation was achieved, as recurring themes emerged, and participants provided rich and focused reflections on their VR experiences. Telephone interviews were chosen to minimize participant burden and allow patients to respond from the comfort of their own homes. Although nonverbal cues were not captured, previous studies suggest that telephone interviews can obtain equally rich verbal data and enhance openness in sensitive contexts [26,38]. Prior experience with TKA surgery provides patients with a basis for reflection, which is essential for how they assimilate information delivered via VR. However, as information needs are highly individual, this is not considered to have affected the study’s credibility [32]. An equal gender distribution ensured that the perceptions and experiences of both genders were reflected. However, the present study did not stratify experiences by gender, age, or educational level. Future research could explore whether these demographic factors influence how patients perceive and benefit from VR-based information. The technical issues that required patients to move their heads to follow the presentation may have affected the study’s credibility [32]. However, these problems were correctable, highlighting the importance of providing prompt technical support. Considering the rapid development of VR technology, the dependability of this study is strengthened by its short data collection period [32]. The fact that two researchers conducted the interviews may have influenced the dependability [32]. To minimize this risk, the interviews were conducted consistently, using a standardized interview guide.

The confirmability is strengthened by discussions of the findings among all authors during data analysis [32]. Any discrepancies were resolved through dialogue and re-examination of the transcripts. Furthermore, a clear description of the interviews and analysis is provided, with the results presented and illustrated with quotations to facilitate readers’ assessment of the study’s transferability [32].

## 6. Conclusions

Information delivered through immersive VR improved both the subjective and objective understanding of the perioperative process and promoted conditions for patient involvement in their own care. VR technology has the potential to meet individual information needs. The challenges lie in ensuring that technical solutions function as intended and in optimizing the conditions for delivering information according to individual needs. Technology, however, cannot replace the information provided by healthcare professionals; rather, it should be regarded as a valuable complement that enhances, but does not substitute for, human interaction and clinical expertise.

Moreover, the optimal timing for delivering preoperative information requires further investigation to determine when it yields the most significant benefit within the perioperative process. Future research should also address the effects of VR-based information on patients’ anxiety and preoperative worry.

## Figures and Tables

**Figure 1 healthcare-13-02896-f001:**
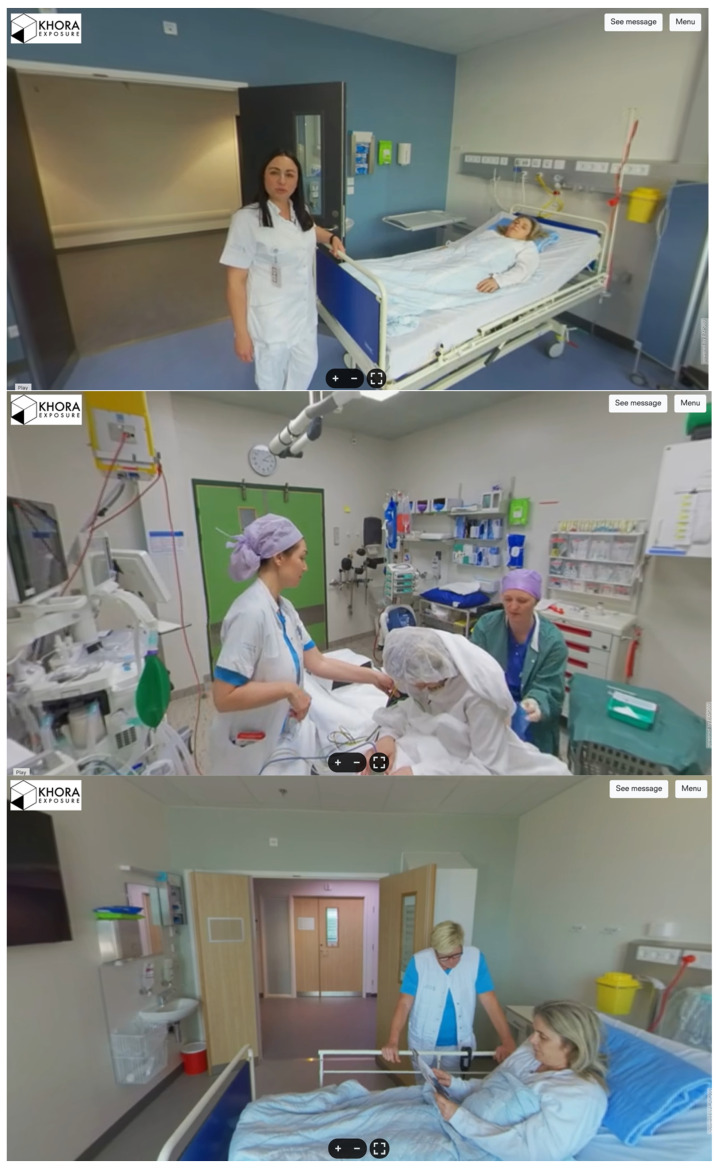
Screenshots of VR content.

**Table 1 healthcare-13-02896-t001:** Characteristics of Study Participants (*n* = 14).

Variable	Median (IQR)	*n*
Age, years	74 (61–82)	
Gender		
Male/Female		6/8
Education level		
Undergraduate		3
Upper secondary school		11
Previous surgery experience		
Yes		10
No		4

**Table 2 healthcare-13-02896-t002:** Examples from data analysis.

Meaning Unit	Condensed Meaning Unit	Code	Subcategory	Category	Mean Theme
Of course, I thought it was strange with those glasses, but I got used to them very quickly.	Strange at first, but I quickly got used to the glasses	Getting accustomed to VR glasses	Adapting to VR glasses.	Using VR as an information tool	Individualized VR information enhances the patient experience.
It was a better way. It’s more detailed and you can understand it better.	More detailed and understandable information	Insight	Take part in an immersive experience		
There weren’t just those three people in the video around her. I walk into a room filled with many people.	The video shows the wrong number of staff.	Pre-Information difference	Discrepancies between VR and clinical experience are annoying	Apply the information provided	
I knew what was going to happen, now that I was in the hospital, and they started getting me ready for that spinal anesthesia. I really enjoyed it.	Confirmation of what was going to happen.	Contributed to understanding the process	Influence on expectations and compliance		

**Table 3 healthcare-13-02896-t003:** Overview of the results.

Main Theme	
Individualized VR Information with VR means additional value to the patient experience	
Categories	Subcategories
Using VR as an information tool	Adapting to VR grasses
	Take part in an immersive experience
Apply the information provided	Discrepancies between VR and clinical experience are annoying
	Influence on expectations and compliance

## Data Availability

The qualitative data on which this analysis was conducted are not publicly available due to ethical concerns regarding the confidentiality of participants. Further, consent was not obtained from participants to share information from interview transcripts with third parties not involved in the research, and the ethical approval for this study does not permit the sharing of such information.

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
