# Peer review of "The Virtual Reality Tour: Immersive Preoperative Information for Elderly Patients"

_healthcare, 2025, doi:10.3390/healthcare13222896_

Round 1
Reviewer 1 Report
Comments and Suggestions for Authors
- Although the paper emphasizes the application of VR in preoperative education for elderly patients, it does not sufficiently highlight its differentiated contribution compared to existing literature. It is recommended to clarify more explicitly in the Introduction or Discussion which research gaps this study addresses and in what aspects its novelty is reflected.
- The average interview duration was approximately 10 minutes, which may be relatively short and could limit the depth of exploration into patients’ experiences. It is recommended to acknowledge this in the methodological limitations and explain how the researchers ensured that sufficient information was obtained.
- Emphasis on clinical significance: It is recommended to highlight more clearly in the discussion the clinical implications of the study results, particularly how they may influence preoperative education and overall care for elderly patients.
- The study did not examine the potential impact of factors such as gender, age group, or education level on the VR experience. It is recommended to address this point in the Results or Discussion section, at least as a direction for future research, to demonstrate consideration of sample heterogeneity.
Author Response
Dear Reviewer 1
We appreciate your comments. Please see the attachment.

Reviewer 2 Report
Comments and Suggestions for Authors
The idea presented in the article is interesting and can provide a great deal of information about the behavior of elderly patients. The interest is to see if virtual reality is a useful tool to help improve the emotional state of patients before surgery and indirectly afterward. A very interesting topic.
However, the article presented suffers from many structural difficulties. Overall, the article does not follow scientific standards for evaluation. Many sections suffer from vague and difficult-to-understand descriptions. There are articles in the literature that address similar issues and detail the evaluation protocol in detail.
I would like to offer you some objective criticisms:
- We cannot judge the visual quality, realism, or relevance of the content. It is difficult to imagine the experimental setup.
- We do not know exactly what the patients saw.
- Scientific non-reproducibility: Another researcher could not repeat the described VR experience.
- Lack of methodological transparency: What design choices were made (camera angles, lighting, etc.)? How is the narrative structured?
"Authors must provide visual documentation of the VR content (screenshots, storyboards) and equipment setup to allow for an adequate assessment of the quality and ecological validity of the intervention. Without these essential visual aids, the scientific rigor and reproducibility of the study cannot be assessed."
- Section 2.1: Study Design
Why qualitative? No justification for the methodological choice over a quantitative approach (e.g., measuring anxiety using validated scales). The description is too generic. We know which method was used, but not how rigorously it was applied.
For subjective analysis, standard questionnaires exist such as the USE, SUS, NASA-TLX, etc.
Section 2.3: Data Collection
Why telephone interviews and not face-to-face interviews? This affects data quality (no nonverbal language).
How was "saturation" defined and achieved? It is a key concept in qualitative, but it is not operationalized.
English could be improved
Author Response
Dear Reviewer 2
We appreciate your comments. Please see the attachment.

Reviewer 3 Report
Comments and Suggestions for Authors
Overall, a very interesting read! The study is timely and relevant, the manuscript is well-structured, the methodology is generally sound, and the discussion is anchored in the context of existing literature. Some specific comments follow:
Major points:
- How were the patients recruited/chosen? Was there a random selection from the participants of the randomized trial? Why 14? The methods section states an inclusion criterion of "aged 18 years or older", but the manuscript's title focus on “elderly Patients". The recruited sample's age range starts at 57…
- Telephone interviews lasted a median of 10.5 minutes, with a range of 6.5–16 minutes. This is a remarkably short duration for semi-structured interviews intended to explore patient experiences in depth. While the authors state that data saturation was achieved, this claim is difficult to substantiate given the limited time spent with each participant. The authors should elaborate in the "Methodological Considerations" section on how they assessed saturation and address the potential limitation that the short interview duration may have precluded the emergence of deeper, more nuanced themes…
- 10 of the 14 participants had previous surgery experience and the discussion rightly points out that this is a critical factor influencing how patients assimilate new information. However, the results are not stratified or analyzed in a way that allows readers to assess the differences between first-time surgery patients and those with prior experience. Can authors present a more detailed comparative analysis (even if brief) or discussion of themes specific to each group? For instance, was the VR tool more valuable for novice patients? Did it primarily serve as a "refresher" or "confirmation" for experienced patients, as hinted in the results?
Minor points
- Authors should be more explicit in declaring that this qualitative study was nested within the intervention arm of a larger randomized controlled trial and also provide a summary description of the trial. It’s important that readers are made clear at the start that the current study's aim was not to compare the two groups, but to conduct an exploration of the experiences of patients who received the VR intervention.
- Participant quotes in the results section are attributed using transcript page numbers (e.g., "(p.10)"). This is not meaningful for the reader. The quotes should be anonymized and consistently attributed to a specific participant (e.g., "Participant 4," "P4," etc.) to allow the reader to track perspectives, if necessary, and to confirm that quotes are drawn from a range of participants.
- Please, state that the final transcripts were fully verified against the audio recordings by the researchers (as is implied).
- The table presenting participant characteristics could be improved. The "Age, years" row could include the mean and standard deviation in addition to the range.
- Table 3 reads "Individualized VR Information with VR means additional value..." , while the text reads "Individualized VR information adds value...". Please ensure consistency.
- The Results section makes a definitive negative claim: "None of the patient's felt that it affected their experience of pain during the perioperative period". Was this a specific question in the interview guide, or is this an inference from the absence of comments on pain? Please clarify how this conclusion was reached.
- In the references, there are some inconsistencies and language-specific artifacts. For example, reference 8 includes "[henvist 12. maj 2025]" which we can infer are Danish terms… Please review the list and formatting.
Author Response
Dear Reviewer 3
We appreciate your comments. Please see the attachment.

Round 2
Reviewer 1 Report
Comments and Suggestions for Authors
- Although the title and abstract are clear, the abstract is somewhat narrative in style and lacks emphasis on the study’s significance and innovation. It is recommended to add a concluding statement at the end of the abstract to clearly highlight the scientific contribution of this study.
- The conclusion mentions that VR can benefit individuals with low health literacy, cognitive impairment, or language barriers; however, these groups were not included in the study sample. It is recommended to express this statement more cautiously.
- There are minor grammatical and spelling errors, and some sentences are slightly colloquial. It is recommended to further polish the English language throughout the paper.
Reviewer 2 Report
Comments and Suggestions for Authors
One significant issue that was not addressed in the paper is identified in Section "4. Results." Regarding user questionnaires, authors didn't show any results.
Typically, we use a "5-likert" or "7-likert" scale and build graphs to get more information about the usefulness and usability of virtual reality for elderly patients. This study should be included in the paper.
Comments on the Quality of English LanguageEnglish could be improved
